# The Effects of Barbed Repositioning Pharyngoplasty in Positional and Non-Positional OSA Patients: A Retrospective Analysis

**DOI:** 10.3390/jcm11226749

**Published:** 2022-11-15

**Authors:** Giovanni Cammaroto, Claudio Moretti, Giuseppe Di Prinzio, Isotta Campomagnani, Giannicola Iannella, Angelo Cannavicci, Giuseppe Meccariello, Andrea De Vito, Antonino Maniaci, Jerome Renè Lechien, Carlos Chiesa-Estomba, Christian Calvo-Henriquez, Paula Martinez Ruiz de Apodaca, Marina Carrasco Llatas, Ahmed Yassin Bahgat, Guillermo Plaza, Carlos O’Connor-Reina, Luca Cerritelli, Virginia Corazzi, Chiara Bianchini, Andrea Ciorba, Stefano Pelucchi, Claudio Vicini

**Affiliations:** 1Ent Department, Morgagni Pierantoni Hospital, 47121 Forlì, Italy; 2Sleep Surgery Group, Young-Otolaryngologists of the International Federations of Otorhino-Laryngological Societies (YO-IFOS), 13005 Marseille, France; 3Ent Department, University of Ferrara, 44121 Ferrara, Italy; 4Department of Sensory Organs, “Sapienza” University of Rome, 00100 Rome, Italy; 5Otolaryngology and Head-Neck Surgery Unit, Department of Head-Neck Surgeries, Santa Maria delle Croci, Azienda USL della Romagna, 48121 Ravenna, Italy; 6Department of Medical, Surgical and Advanced Technologies G.F. Ingrassia, ENT Section, University of Catania, Via Santa Sofia, 95100 Catania, Italy; 7Department of Human Anatomy and Experimental Oncology, Faculty of Medicine, UMONS Research Institute for Health Sciences and Technology, University of Mons (UMons), 7000 Mons, Belgium; 8Department of Otorhinolaryngology-Head and Neck Surgery, Hospital Universitario Donostia, Biodonostia Health Research Institute, 20014 San Sebastian, Spain; 9Department of Otolaryngology, Hospital Complex of Santiago de Compostela, 15706 Santiago de Compostela, Spain; 10Department of Otorhinolaryngology, Dr. Peset University Hospital, 46017 Valencia, Spain; 11Department of Otorhinolaryngology, Alexandria University, Alexandria 21928, Egypt; 12Otorhinolaryngology Department, Hospital Universitario de Fuenlabrada and Hospital Sanitas la Zarzuela, Universidad Rey Juan Carlos, 28042 Madrid, Spain; 13Department of Otorhinolaryngology, Hospital Quironsalud Marbella, 29603 Marbella, Spain

**Keywords:** OSA, pharyngoplasty, positional OSA, sleep surgery

## Abstract

PURPOSE: The aim of our retrospective study is evaluating the effectiveness of barbed repositioning pharyngoplasty (BRP) in a consecutive cohort of patients and assessing its impact on positional indexes in order to potentially identify specific obstructive sleep apnea (OSA) phenotypes for patients who might benefit more significantly from this intervention. METHODS: A single-center retrospective study with baseline and follow-up type III sleep tests evaluating the Apnea Hypopnea Index (AHI), supine AHI, non-supine AHI, oxygen desaturation index (ODI), mean SaO2, percentage of time spent at SaO2 below 90% (CT90), and lowest oxygen saturation (LOS) were performed. The patients were then divided into groups according to Sher’s criteria and Amsterdam Positional OSA Classification (APOC). Parametric and non-parametric tests and univariate and multivariate analyses were conducted. RESULTS: The study finally included 47 patients. The statistical analysis showed significant improvement in AHI, supine AHI, non-supine AHI, and ODI after surgery. The linear regression showed that high values of baseline AHI, AHI supine, and AHI non supine predict more significant postoperative reductions in AHI, AHI supine, and AHI non supine, respectively. Therapeutic success was achieved in 22 patients out of 47. The logistic regression did not find any independent risk factors for success. The most significant reduction in AHI, supine AHI, and non-supine AHI was observed in the APOC 3 group while the APOC 1 patients experience a substantially lower improvement. CONCLUSIONS: BRP appears to be an effective surgical procedure for the treatment of OSA. The non-positional patients might benefit more from BRP in comparison with positional patients. Moreover, OSA severity should not be considered an absolute contra-indication for this surgical procedure.

## 1. Introduction

The interest of the scientific community in studying the correlation between body position during sleep and quality of sleep itself has its beginnings in the 1980s with De Koninck et al. These authors assumed that what distinguished “good sleepers” from “bad sleepers” was the amount of time spent in the supine position, indicated as generally associated with the development of sleep breathing disorders [1]. This intuition, based on the quality of sleep subjectively perceived by patients, was objectified through the polysomnographic studies of Cartwright, who introduced the concept of the “positional patient” (PP) [2]. The author first proposed the distinction between the PP and non-positional patient (NPP), establishing the arbitrary criteria of a difference of at least 50% between the supine and non-supine Apnea Hypopnea Index (AHI) value.

To date, there are no universally recognized criteria of Positional Obstructive Sleep Apnea (POSA); nevertheless, the most recent and used classification is the APOC system (Amsterdam Positional Osa Criteria). This classification aims to distinguish between patients who could benefit or not from positional therapy (PT) [3]. 

According to the APOC classification, approximately 56–75% of patients can be defined as POSA [4]; moreover, POSA patients appear to suffer from less severe OSAS in comparison with non-POSA patients [5]. More recent studies have also estimated the prevalence of positionality in patients >65 yo at 66.9% [6] and in patients <18 yo at 18.9% [7].

From a pathophysiological point of view, there are several factors to take into consideration in the pathogenesis of POSA: in a 2014 article, Joosten et al. include a set of elements that contribute to greater collapse of the upper airways (UA) in the supine position and to the genesis of POSA:-Force of gravity;-Critical pressure of pharyngeal closure (PCrit), which increases in the supine position compared to the lateral position;-Pulmonary volumes decrease in the supine position;-Decreased tone of the genioglossus muscle during sleep [8].

PT appears to be an interesting alternative to C-PAP for the treatment of POSA considering its low costs, simplicity, and higher compliance when compared to ventilatory therapy [9,10]. However, several systematic reviews have shown that Continuous-Positive Airway Pressure therapy (C-PAP) remains more effective in the treatment of these patients when compared with PT [11]; despite this, PT might represent the first line treatment for APOC-1 patients [12].

Regarding surgery, in recent decades, several palatal interventions have been proposed; among these, the most known are uvulopalatopharyngoplasty (UPPP), expansion sphincter pharyngoplasty (ESP), and barbed repositioning pharyngoplasty (BRP). Overall, BRP and ESP were found to be most effective both as a single-modal treatment [13] and as part of a trans-oral-robotic-associated multimodal treatment for tongue base and hypopharynx/supraglottic surgery [14]. Although these evidence of efficacy in NPP patient are strong, in the literature there is a lack of studies that focus primarily on PP and its relationship with surgery. Velopharyngeal surgery has been shown to have some degree of efficacy [15,16], but prognostic indices of success and efficacy of BRP have not yet been studied.

The aim of our retrospective study is evaluating the effectiveness of BRP in a consecutive cohort of patients and assessing its impact on positional indexes in order to potentially identify specific Obstructive Sleep Apnea (OSA) phenotypes for patients who might benefit more significantly from this intervention.

## 2. Materials and Methods

### 2.1. Study Design

Single-center retrospective study: The medical charts of consecutive OSA patients who underwent BRP, tonsillectomy, and septo-turbinoplasty between April 2014 and June 2021 were evaluated retrospectively. The BRP was performed as previously described [13,14].

### 2.2. Inclusion Criteria

Patients suffering from mild to severe OSA (AHI ≥ 5 events/h) with a certain degree of nasal obstruction;Grades 1–2 tonsillar hypertrophy;Aged between 18 and 65 years old;BMI ≤ 35.

### 2.3. Exclusion Criteria

Serious psychiatric, cardiopulmonary, or neurological disease;Previous tonsillectomy and OSA surgery;Significant craniofacial anomalies and or/narrow upper maxilla;Pharmacological treatment for the OSA or drugs with an impact on the cognitive function;Grades 3–4 tonsillar hypertrophy;Follow-up < 6 months and >12 months;Preoperative and Postoperative sleep tests with supine time less than 25% or more than 75% of total sleep time;Central or mixed apnea events >25% at preoperative sleep test.

The following clinical data were collected: age, gender, body mass index (BMI), full medical history, and upper airways examination.

All patients underwent preoperative and postoperative (between 6 and 12 months after surgery) home sleep apnea tests (HSAT), Epworth sleepiness score test (ESS), and BMI evaluation.

Baseline and follow-up type III sleep tests evaluating the AHI, supine AHI, non-supine AHI, oxygen desaturation index (ODI), mean SaO2, percentage of time spent at SaO2 below 90% (CT90), and lowest oxygen saturation (LOS) were performed.

All the sleep studies were carried out in an unattended way by means of a Polymesam Unattended Device 8-channel, reviewed, and scored by the same expert in sleep medicine according to the American Academy of Sleep Medicine Guidelines [17].

Therapeutic success was defined according to Sher’s criteria: achievement of a postoperative value of AHI < 20 and a 50% improvement in the preoperative AHI value [18].

A comparative analysis between preoperative and postoperative variables was performed.

Delta AHI (preoperative AHI—postoperative AHI), Delta AHI supine (preoperative AHI supine—postoperative AHI supine), and Delta AHI non-supine (postoperative AHI non supine—preoperative AHI non supine) were also calculated in order to favor statistical analysis.

The patients were then divided into groups according to Sher’s criteria and Amsterdam Positional OSA Classification (APOC) [3].

(1)APOC 1: AHI value in the best sleeping position (BSP) is <5; patients can be treated with PT.(2)APOC 2: severity of OSA in the BSP is one degree lower in comparison with total AHI. These patients might benefit from PT without, however, being cured completely.(3)APOC 3: total AHI > 40 and AHI in the BSP reduced by at least 25%; these patients cannot be cured by exclusive PT. PT might however increase compliance to other therapeutic options.(4)Not classifiable group.

*Main outcome:* To compare the effectiveness of lateral pharyngeal walls and nasal surgery between positional and non-positional OSA patients.

*Secondary outcomes:* To evaluate the effectiveness of lateral pharyngeal and nasal surgery in OSA patients, to identify potential prognostic factors, and to investigate the differences between failures and successes.

Local ethics committees or institutional review boards approved the study.

For this type of study formal consent was not required.

### 2.4. Statistical Analysis

To test the differences between the paired and unpaired groups, parametric (*t*-test) and non-parametric (Kruskal–Wallis test) tests were used as appropriate. The role of each factor (univariate analysis) and their independent effect (multivariate analysis) were explored using logistic or linear regression models as appropriate. The surgical success was set as the dependent value for logistic regression. The probability values lower than 0.05 were considered as statistically significant. All analyses were performed using SPSS version 20 (IBM Corporation, Armonk, NY, USA).

The Local Ethics Committee approved the study (Ref. Number 4842) on 21 March 2022.

## 3. Results

The medical chart research filtered with inclusion and exclusion criteria retrieved 213 cases. The patients who did not undergo a post-operative sleep test, who were not present at the follow up, and the patients who did not accept to be part of the study (166) were excluded.

The study finally included 47 patients. The mean age at surgery was 54.5 (SD 10.41).

All these preoperative and postoperative data are shown in Table 1.

The statistical analysis showed a significant improvement in AHI, supine AHI, non-supine AHI, and ODI after surgery (Table 1).

In Figure 1, the postoperative change of AHI, supine AHI, and non-supine AHI is well demonstrated.

A linear regression was performed to test the relationship between baseline AHI, AHI supine, and AHI non supine with delta-AHI, delta AHI supine, and delta AHI non supine, respectively. The results showed that high values of baseline AHI, AHI supine, and AHI non supine predict more significant postoperative reductions in AHI, AHI supine, and AHI non supine, respectively (Figure 2 shows the linear regression for the baseline AHI and delta-AHI; R 0.815, *p* < 0.001).

According to Sher’s criteria, the therapeutic success was achieved in 22 patients out of 47. The PPs presented with the worst therapeutic outcomes with a success rate standing at around 22%, while the APOC 3 patients reported 60% success. Moreover, four patients reported a significant worsening of post-operative AHI (>20 events per hour in all four patients), possibly related to a consistent increase in BMI (>5 points of BMI in all four patients).

The comparative analysis of polysomnographic parameters in the responders and non-responders groups highlighted significantly higher deltaAHI, delta supine AHI, and delta non-supine AHI in responders (Table 2). Moreover, the lower preoperative average spO2 values and higher preoperative AHI non supine figures were observed in responders. The logistic regression did not find any independent risk factor for success (Table 2).

The patients were then divided into four groups according to the Amsterdam Positional OSA Classification (APOC): APOC 1 (N = 9), APOC 2 (N = 24), APOC 3 (N = 9), and APOCN (not classifiable group: in this group we included those patients who did not meet the criteria of the other groups; N = 5). The most significant reduction in AHI, supine AHI, and non-supine AHI was observed in the APOC 3 group, while the APOC 1 patients experienced a substantially lower improvement (Table 3, Figure 3, Figure 4 and Figure 5).

## 4. Discussion

Few studies primarily focusing on the effectiveness of velopharyngeal surgery for PPs have been published in the literature. On the other hand, several papers have shown the effectiveness of lateral pharyngeal wall techniques for the treatment of OSA patients [19,20,21].

The shift from respective procedures such as UPPP to more conservative lateral pharyngoplasties has certainly led to a concrete improvement of therapeutic efficacy associated with a significant decrease in morbidity [13].

However, the research community is still facing titanic challenges in depicting accountable predictors of surgical failures.

UA surgery appears to be an effective tool in managing pharyngeal collapsibility; on the other hand, arousal threshold, muscular responsiveness and loop gain do not seem the main targets of this therapeutic strategy.

Schwartz et al. recorded a significant reduction in Pcrit in patients experiencing satisfactory responses to UPPP. However, these authors did not manage to find reliable predictors in their series [22].

Conversely, Joosten et al. found out that patients reporting better outcomes after multilevel surgery presented a lower preoperative loop gain [23].

Taking into account the complexity of sleep laboratory analysis, phenotyping OSA patients appears to be possible also in an outpatient setting. 

For instance, the evaluation of the tonsil grade, Friedman palate position, and Friedman lingual tonsil grade might help in identifying patients in whom single-level velopharyngeal surgery should be avoided [24].

Moreover, the adoption of drug induced sleep endoscopy may allow an increase in the efficacy of velopharyngeal techniques, as observed by several authors. Furthermore, certain collapse patterns, such as complete circular velar collapse, seem to be not properly treated with a respective technique, while it might be correctly addressed by means of a lateral pharyngoplasty procedure [25,26,27].

In our series, a statistically significant improvement of sleep test parameters was observed post-operatively. However, according to Sher’s criteria, the responders’ rate stood around 47%, lower than reported by previous studies. This outcome might be related to some strict exclusion criteria such as tonsillar hypertrophy and supine time cut-off of sleep tests. Moreover, patients included in this series did not undergo preoperative drug-induced sleep endoscopy, thus leading to a potentially less accurate candidate selection. Finally, few patients reported also a significant increase in BMI with a consistent worsening of post-operative AHI.

The linear regression put in evidence that high values of baseline AHI, supine AHI, non-supine AHI, respectively, predict more significant postoperative reductions in AHI, supine AHI, non-supine AHI, respectively: from this analysis, OSA severity does not appear to be a contraindication of palate surgery. 

The logistic regression comparing responders with non-responders did not highlight any independent risk factors. The limited sample size might explain the above outcome.

Further comparative analysis between failures and successes demonstrated a higher preoperative non-supine AHI and lower preoperative mean SpO2 in the responders group. 

When applying APOC classification to our series, some interesting figures emerged from the analysis.

In particular, APOC 2 and especially APOC 3 patients experienced the most significant improvement of AHI, non-supine AHI, and, surprisingly, also supine AHI (APOC 3 success rate 60%).

On the other hand, pure PPs presented with the worst therapeutic outcomes with a success rate standing around 22%.

The main reason of these findings is to be found in the rationale of the BRP technique. This procedure aims to stabilize the lateral pharyngeal walls and is not addressed to correct oral tongue or base of tongue collapses, acting neither in reducing anatomic volumes nor on favoring genioglossus activation.

These outcomes might suggest that palate surgeries might be indicated for NPP, as reported by other research groups [16,28].

PT and oral appliances can be considered valuable options for multimodal treatment in case of the failure or partial success of palate surgery. In fact, treatment successes of mandibular advancement devices do not seem to be influenced by position-dependent obstructive sleep apnea and might be effective in improving supine AHI [29].

Taking into account the limited sample size and the retrospective nature of this study, the following consideration can be proposed. BRP appears an effective surgical procedure for the treatment of OSA. NPP might benefit more from BRP in comparison with PPs. Moreover, OSA severity should not be considered as an absolute contraindication for this surgical procedure. This study shows the importance of focusing on more than one single polysomnography parameter and the need for evaluating positional indexes in the process of selecting candidates for pharyngoplasty.

In conclusion, further studies with longer follow-ups, larger series, and prospective designs are needed in order to better identify surgical candidates that might benefit more from BRP. However, pharyngeal surgery appears a promising and viable option for the treatment of certain specific subsets of patients.

## Figures and Tables

**Figure 1 jcm-11-06749-f001:**
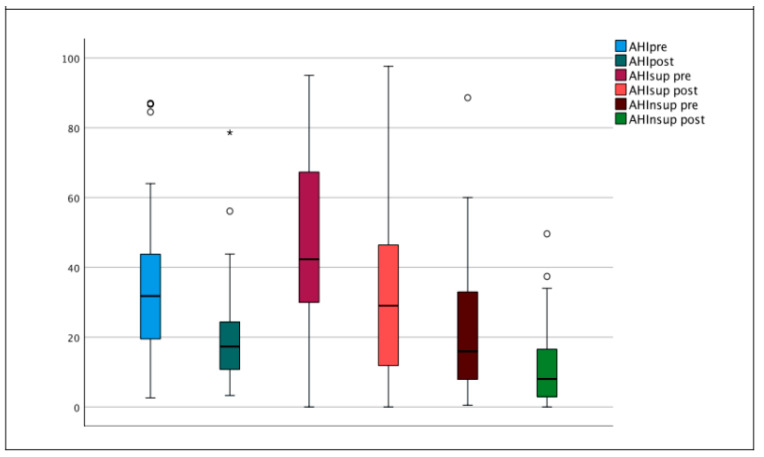
Boxplot showing postoperative improvement of AHI, AHI supine, and AHI non supine. AHI (Apnea Hypopnea Index), AHI sup (Apnea Hypopnea Index supine), AHI nsup (Apnea Hypopnea Index non-supine). Pre (Pre-operative), Post (Post-operative). Asterisk and circles represent outliers.

**Figure 2 jcm-11-06749-f002:**
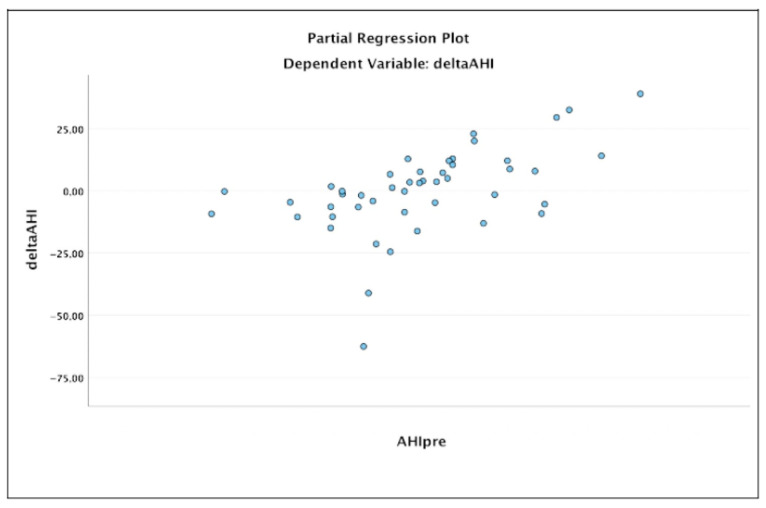
Linear regression graphs showing the relationship between preoperative AHI and delta-AHI.

**Figure 3 jcm-11-06749-f003:**
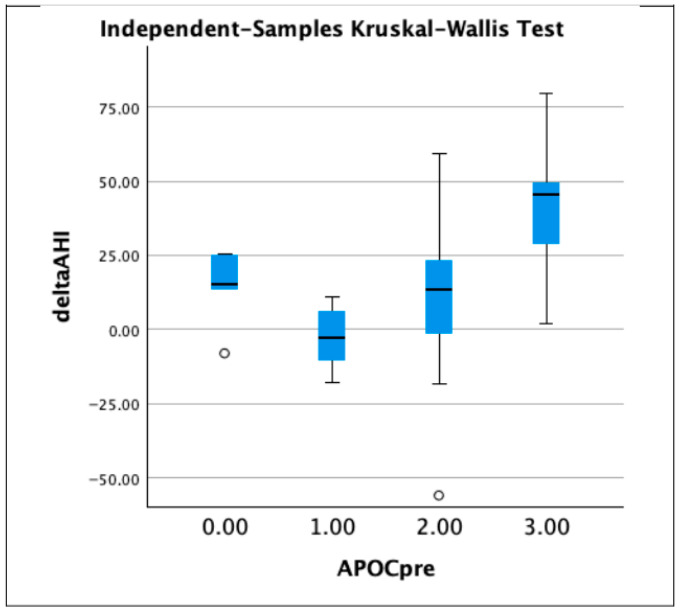
Boxplots showing delta AHI in patients divided in groups according to APOC classification (group 0 stands for non-classifiable APOC; 1.00 = APOC 1, 2.00 = APOG 2, and 3.00 = APOC 3). Circle represent outliers.

**Figure 4 jcm-11-06749-f004:**
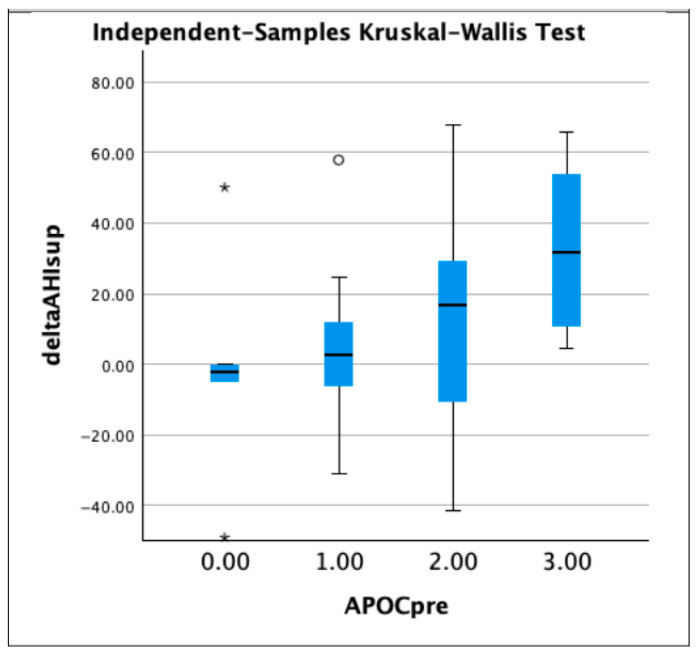
Boxplots showing delta AHI supine in patients divided in groups according to APOC classification (group 0 stands for non-classifiable APOC; 1.00 = APOC 1, 2.00 = APOG 2, and 3.00 = APOC 3). Asterisks and circle represent outliers.

**Figure 5 jcm-11-06749-f005:**
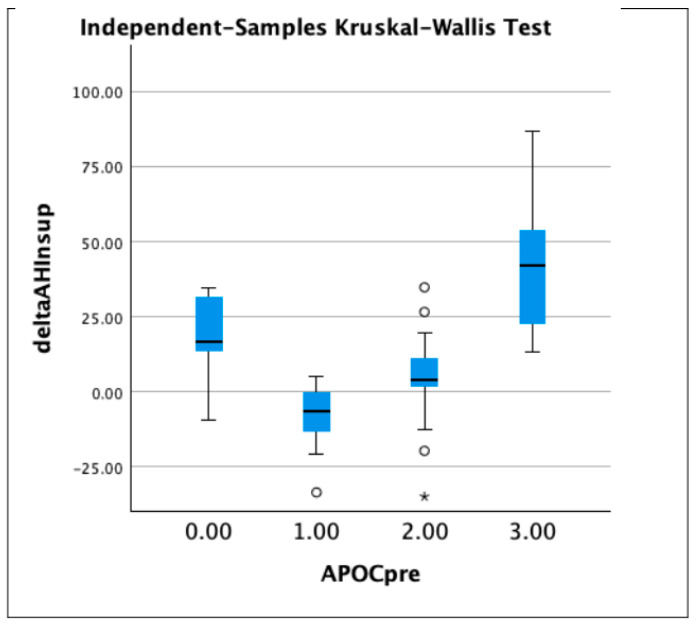
Boxplots showing delta AHI non supine in patients divided in groups according to APOC classification (group 0 stands for non-classifiable APOC; 1.00 = APOC 1, 2.00 = APOG 2, and 3.00 = APOC 3). Asterisk and circles represent outliers.

**Table 1 jcm-11-06749-t001:** Preoperative and postoperative data of our series (delta = preoperative–postoperative values) and paired *t*-test between preoperative and postoperative data are shown.

N° of Patients = 47	Before Surgery		After Surgery		Delta		
	Mean	Std. Deviation	Mean	Std. Deviation	Mean	Std. Deviation	Two-Sided *p*
BMI	27.8	3.80	27.6	3.81	−0.05	5.31	0.952
ESS	10.3	5.37	6.9	5.43	2.25	3.28	0.094
AHI	35.4	20.10	20.4	14.72	14.9	25.18	**<0.001**
AHI sup	45.7	25.53	32.5	22.61	13.2	30.33	**0.004**
AHI nsup	22.3	19.62	11.8	11.18	10.5	23.12	**0.003**
ODI	31.8	20.44	21.0	15.43	10.3	26.09	**0.022**
AvgSpO2	93.0	1.74	93.7	1.93	−0.41	2.92	0.538
LOS	76.2	14.71	77.3	17.26	1.7	18.33	0.660
CT90	15.0	16.35	10.4	16.39	4.3	20.81	0.357

BMI (body mass index), ESS (Epworth scale score), AHI (Apnea Hypopnea index), AHI sup (Apnea Hypopnea Index supine), AHI nsup (Apnea Hypopnea Index non-supine), ODI (Oxygen Desaturation Index), AvgSpO2 (Average Saturation), LOS (Lowest Oxygen Saturation), CT90 (Percentage time with saturation below 90%).

**Table 2 jcm-11-06749-t002:** Comparative analysis between responders and non-responders (Independent *t*-test). Logistic regression to test the influence of preoperative parameters on success is also shown.

	Responders (tot = 22)		Non-Responders (tot = 25)		
	Mean	Std. Dev.	Mean	Std. Dev.	Significance Two-Sided *p*
deltaAHI	29.85	18.86	1.81	22.82	**<0.001**
deltaAHI sup	25.84	29.42	2.11	27.04	**0.006**
deltaAHI nsup	23.51	22.27	−0.94	17.28	**<0.001**
AgeAtSurgery	54.14	11.90	54.89	9.139	0.810
BMI pre	28.14	4.02	27.58	3.65	0.618
ESS pre	10.23	5.02	10.33	5.94	0.618
AHI pre	40.48	18.85	30.87	20.44	0.102
AHIsup pre	49.87	26.01	42.12	25.07	0.304
AHInsup pre	28.11	22.44	17.27	15.49	**0.058**
ODI pre	35.20	17.64	28.75	22.65	0.313
AvgSpO2 pre	91.83	2.05	93.55	1.40	0.022
LOS pre	78.53	6.84	74.36	18.83	0.393
ct90 pre	18.60	16.42	12.36	16.30	0.327
**Logistic Regression**					
		**B**		**Odds Ratio**	**Sig.**
AHIpre		0.002		1.002	0.942
AHIsuppre		0.008		1.008	0.649
AHInsuppre		0.032		1.032	0.238
AgeAtSurgery		−0.006		0.994	0.840
BMIpre		−0.057		945	0.581
Constant		0.632		1.882	0.826

**Table 3 jcm-11-06749-t003:** Kruskal–Wallis test comparing delta AHI, delta AHI supine, and delta AHI non supine in APOC groups (APOC N stands for non-classifiable APOC).

Delta AHI			
	Z score	Std. Error	Sig.
APOC N VS APOC3	−11.400	7.648	0.136
APOC1 VS APOCN	13.378	7.648	0.080
APOC1 VS APOC2	−10.986	5.359	**0.040**
APOC1 VS APOC3	−24.778	6.463	**<0.001**
APOC2 VS APOCN	2.392	6.740	0.723
APOC2 VS APOC3	−13.792	5.359	**0.010**
**Delta AHI Supine**			
	**Z score**	**Std. Error**	**Sig.**
APOCN vs. APOC1	−2.022	7.648	0.791
APOCN vs. APOC1	−6.258	6.740	0.353
APOCN vs. APOC1	−16.800	7.648	**0.028**
APOC1 vs. APOC2	−4.236	5.359	0.429
APOC1 vs. APOC3	−14.778	6.464	**0.022**
APOC2 vs. APOC3	−10.542	5.359	**0.049**
**Delta AHI Non-Supine**			
	**Z score**	**Std. Error**	**Sig.**
APOC N vs. APOC3	−9.478	7.647	0.215
APOC1 vs. APOCN	20.356	7.647	**0.008**
APOC1 vs. APOC2	−11.118	5.359	**0.038**
APOC1 vs. APOC3	−29.833	6.463	**<0.001**
APOC2 vs. APOCN	9.238	6.740	0.171
APOC2 vs. APOC3	−18.715	5.359	**<0.001**

## Data Availability

The datasets generated during and/or analyzed during the current study are available from the corresponding author on reasonable request.

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
