# Peer review of "The Effects of Barbed Repositioning Pharyngoplasty in Positional and Non-Positional OSA Patients: A Retrospective Analysis"

_jcm, 2022, doi:10.3390/jcm11226749_

Round 1

Reviewer 1 Report

1.     Space and paragraphs need to be adjusted.

2.     Citation and reference format needs to be uniform.

3.     Please provide IRB approval number and date.

4.     If abbreviations appear at first time, please provide full name in the article.

5.     Flowcharts or graphic abstracts are more informative about the study design.

6.     Tables and figures need to be strengthened in design.

7.     The titles of tables and figures need to be concise.

8.     Conclusion needs to be elaborated and powerful.

9.     The title of the abbreviation section should be added.

Author Response

  1. Space and paragraphs need to be adjusted.

Response: Thanks for your comments, space and paragraphs has been adjusted.

  1. Citation and reference format needs to be uniform.

Response: Thanks for your comments, we have kept citations and reference uniform.

  1. Please provide IRB approval number and date.

Response: Thanks for your comments, we have added IRB Statement. Local ethics committees or institutional review boards approved the study. For this type of study formal consent was not required.

  1. If abbreviations appear at first time, please provide full name in the article.

Response: Thanks for your comments, we have added the full name of abbreviations.

  1. Flowcharts or graphic abstracts are more informative about the study design.

Response: We appreciate the reviewer’s proposal but we feel that the use of flowcharts and graphic abstract might be more useful in a different study design.

  1. Tables and figures need to be strengthened in design.

Response: I would like to thank the reviewer for the suggestion, the tables were revised, however boxplots and linear regressions cannot be changed due to the characteristics of our statistical software

  1. The titles of tables and figures need to be concise.

Response: We appreciate the reviewer’s proposal but we feel that figure legends need to fully describe the purpose of each figure and table.

  1. Conclusion needs to be elaborated and powerful.

Response: Thanks for your comments, conclusion has been revised.

  1. The title of the abbreviation section should be added.

Response: Thanks for your comments, the title of the abbreviation section was added.

Reviewer 2 Report

 statistical analysis: You are therefore saying that for the parametric

distribution, you used the T-test. (What kind of T-test? Independent T-test? Paired T test?). And you also

you were saying that you use for the non parametric distribution Kruskall Wallis test. However, the t test

consists of 2 variables and Kruskall Wallis tests more than 2 variables. You must therefore specify the

test you use for parameters and nonparametric. In addition, have you tested the distribution? I have not

seen in your paper , Shapiro Wilk test or Kolmogorov Smirnov. Furthermore, you said that you are using

version 2 of SPSS, maybe version 20?

Table 1 has very little aesthetics. You put a mean and a standard deviation (and all your other tests are

not parametric), perhaps you should declare the median and the interquartile of non-parametric

distribution (q1 and q3). The mean and standard deviation in Table 1 for AgeAT Surgery I think is

additional. Perhaps you should just write a report into the text. The number of patients has to be below

the age of surgery. There is no Legend in Table 1 for variables. What means ESS, AHI, AHI nsup, CT90?

Explanation of table 1? Where is the number of pacients in Table 1 pre- and post-intervention? What

means delta? The legend should be very clear. The table has to be restructured maybe you want to put

Mean (standard deviation) in parenthesis. What test did you use in table 2? Paired T test? Can you offer

many explanations of table 2? In Table 2, Logistic Regression, you used the B estimate and the

significance. But where are the ODD RATIO and confidence intervals of ODD RATIO? How you

determined by the independent risk factor? What is the dependent variable? At the table 3 you were

showing Z score at Test statistic? Could you provide much information about the table 3? Please provide

the legend of variable in Figure 1. Also, you could specify in the legend the circles (outliers) and the*. At

the figure 2 could you provide some information? Could you describe the value of R and the p value?

Would you mind doing the Kruskal-Wallis boxplot again? Maybe you can insert the variable name

instead of 0,1,2,3? Another mention is that you can make a table of 3 cells and you can place each figure

next to another and is much more enjoyble. The legend for the variables must also be introduced.

Author Response

You are therefore saying that for the parametric distribution, you used the T-test. (What kind of T-test? Independent T-test? Paired T test?).

Response: Thanks for your comments, we specified in table legends.

And you also you were saying that you use for the non parametric distribution Kruskall Wallis test. However, the t test consists of 2 variables and Kruskall Wallis tests more than 2 variables. You must therefore specify the test you use for parameters and nonparametric.

Response: Thanks for your comments, we specified in statistical analysis section.

In addition, have you tested the distribution? I have not seen in your paper , Shapiro Wilk test or Kolmogorov Smirnov.

Response: Thank you for your comment, distribution was tested, however for subgroup analysis non parametric test was preferred being subgroups composed by a limited number of patients.

Furthermore, you said that you are using version 2 of SPSS, maybe version 20?

Response: Thank you for your comment, we have corrected.

Table 1 has very little aesthetics. You put a mean and a standard deviation (and all your other tests are not parametric), perhaps you should declare the median and the interquartile of non-parametric

distribution (q1 and q3).

Response: Thank you for your comment, the only non parametric test is relative to table 3 and figure 3-5. All other tests were parametric.

The mean and standard deviation in Table 1 for AgeAT Surgery I think is additional. Perhaps you should just write a report into the text. The number of patients has to be below the age of surgery.

Response: Thank you for your comment, we revised as suggested.

There is no Legend in Table 1 for variables. What means ESS, AHI, AHI nsup, CT90?

Explanation of table 1? Where is the number of pacients in Table 1 pre- and post-intervention? What means delta? The legend should be very clear.

Response: Thank you for your comment, legend and table were modified as requested.

The table has to be restructured maybe you want to put Mean (standard deviation) in parenthesis.

Response: Thank you for your comment, Mean and SD appear clear.

What test did you use in table 2? Paired T test? Can you offer many explanations of table 2?

Response: Thank you for your comment, we have specified in legend of table 2.

In Table 2, Logistic Regression, you used the B estimate and the significance. But where are the ODD RATIO and confidence intervals of ODD RATIO? How you determined by the independent risk factor? What is the dependent variable?

Response: Thanks for the comments. Dependent variable is success, specified in statistical analysis section. Odd ratio added in the table.

At the table 3 you were showing Z score at Test statistic? Could you provide much information about the table 3?

Response: We appreciate the reviewer’s proposal, table 3 was modified and specified Z score.

Please provide the legend of variable in Figure 1. Also, you could specify in the legend the circles (outliers) and the*.

Response: Thanks for the suggestion, we have modified as required.

At the figure 2 could you provide some information? Could you describe the value of R and the p value?

Response: Thanks for the suggestion, the information was specified in results section.

Would you mind doing the Kruskal-Wallis boxplot again? Maybe you can insert the variable name

instead of 0,1,2,3?

Response: Unfortunately not possible. Legend was improved describing plots in details.

Another mention is that you can make a table of 3 cells and you can place each figure next to another and is much more enjoyble. The legend for the variables must also be introduced.

Response: Thanks for the suggestion, we tried to modified according o the suggestion. But a clearer comprehension of statistical data was not achieved. Therefore, we decided to keep the previous setting.

Round 2

Reviewer 1 Report

1.The changed part needs to provide the updated line number.

2.The titles of tables and figures need to be concise. Relevant supplements can be placed under the table or figure.

Author Response

updated line number: 63-65, 85, 100, 149-150, 178-182, 187, 198, 163, 168, 265, 270, 271, 309, 314, 315, 324-325, 348-351

titles of tables and figures modified as requested. supplements added under tables and figures